# High Dielectric Constant and Dielectric Relaxations in La_2/3_Cu_3_Ti_4_O_12_ Ceramics

**DOI:** 10.3390/ma15134526

**Published:** 2022-06-27

**Authors:** Lei Ni, Chuyi Zhang, Lu Fang

**Affiliations:** School of Materials Science and Engineering, Chang’an University, Xi’an 710064, China; 2019231001@chd.edu.cn (C.Z.); 2020231034@chd.edu.cn (L.F.)

**Keywords:** La_2/3_Cu_3_Ti_4_O_12_ ceramics, high dielectric constant, dielectric relaxation, internal barrier layer capacitor effects

## Abstract

La_2/3_Cu_3_Ti_4_O_12_ ceramics were prepared by the same method of solid-state reaction as CaCu_3_Ti_4_O_12_ ceramics. The structure and dielectric responses for La_2/3_Cu_3_Ti_4_O_12_ and CaCu_3_Ti_4_O_12_ ceramics were systematically investigated by X-ray diffraction, scanning electron microscope, X-ray photoelectron spectroscopy, and impedance analyzer. Compared with CaCu_3_Ti_4_O_12_ ceramics, La_2/3_Cu_3_Ti_4_O_12_ ceramics with higher density and refined grain exhibit a high dielectric constant (ε′ ~ 10^4^) and two dielectric relaxations in a wide temperature range. The dielectric relaxation below 200 K with an activation energy of 0.087 eV in La_2/3_Cu_3_Ti_4_O_12_ ceramics is due to the polyvalent state of Ti^3+^/Ti^4+^ and Cu^+^/Cu^2+^, while the dielectric relaxation above 450 K with higher activation energy (0.596 eV) is due to grain boundary effects. These thermal activated dielectric relaxations with lower activation energy in La_2/3_Cu_3_Ti_4_O_12_ ceramics both move to lower temperatures, which can be associated with the enhanced polyvalent structure in La_2/3_Cu_3_Ti_4_O_12_ ceramics. Such high dielectric constant ceramics are also expected to be applied in capacitors and memory devices.

## 1. Introduction

CaCu_3_Ti_4_O_12_ ceramics have always been of interest in the field of high dielectric constant materials, not only for their high dielectric constant with better temperature and frequency stability but also for their unique dielectric relaxation [1,2,3,4,5,6,7,8,9,10,11,12,13,14]. In the process of exploring the origin of giant dielectric properties and improving the performance of CaCu_3_Ti_4_O_12_ ceramics, it is quite impressive that many perovskite-like ceramics *A*Cu_3_Ti_4_O_12_ (*A* = Cd, Y_2/3_, Sm_2/3_, La_2/3_, Na_0.5_Bi_0.5_, etc.) [9,13,14,15,16,17,18,19,20,21] have been found to have the similar high dielectric constant as CaCu_3_Ti_4_O_12_, which revises the report by Subramanian [2] and brings new questions about the physical origin of dielectric response in *A*Cu_3_Ti_4_O_12_ ceramics. Internal barrier layer capacitor (IBLC) effects related to defect structure in grain and grain boundaries have been widely adopted to explain the origin of high dielectric response in these *A*Cu_3_Ti_4_O_12_ ceramics [10,11,12,13,14,15,16,17,18,19,20,21,22,23,24,25,26,27,28].

La_2/3_Cu_3_Ti_4_O_12_, one of the typical members of the *A*Cu_3_Ti_4_O_12_ family, was reported for the first time with a low dielectric constant of 418 (measured at 100 kHz and 25 °C) [2]. However, Zhang [15] and Yang [19] et al. have shown extremely high dielectric constant (about 10^4^ below 100 kHz) in La_2/3_Cu_3_Ti_4_O_12_ ceramics, which has challenged the former results and immediately stimulated research enthusiasm. Some works [20,21,22,23,24,25,26,27,28] have focused on the effects of fabrication conditions and doping on the structure and properties of La_2/3_Cu_3_Ti_4_O_12_ ceramics. However, there are few comparative studies on the dielectric relaxation and relative mechanism of La_2/3_Cu_3_Ti_4_O_12_ and CaCu_3_Ti_4_O_12_ ceramics with similar sintering parameters, and there are still some questions unsolved, such as, what are the changes in structure, dielectric relaxation caused by the complete substitution of hetero-valent ions on *A* site in *A*Cu_3_Ti_4_O_12_ ceramics. Therefore, it is worthy to comparatively investigate the similarity and difference in structure and dielectric properties in La_2/3_Cu_3_Ti_4_O_12_ and CaCu_3_Ti_4_O_12_ ceramics.

In the present work, La_2/3_Cu_3_Ti_4_O_12_ and CaCu_3_Ti_4_O_12_ ceramics were prepared by the same process. The structure and dielectric properties of La_2/3_Cu_3_Ti_4_O_12_ ceramics were systematically investigated and compared with CaCu_3_Ti_4_O_12_ ceramics. The similarity and difference of dielectric properties and relative mechanism were discussed in detail.

## 2. Materials and Methods

La_2/3_Cu_3_Ti_4_O_12_ were prepared by the solid-state reaction process (as shown in Figure 1) from the high-purity powders of TiO_2_ (99.99%, Aladdin Reagent Co., Ltd., Shanghai, China), CuO (99.9%, Aladdin Reagent Co., Ltd.), and La_2_O_3_ (99.99%, Aladdin Reagent Co., Ltd.). Raw materials were weighed and mixed in a planetary muller for 10 h, were heated twice at 950 °C for 4 h, and then pressed into disks (diameter: 12 mm). The disks were sintered from 1050 °C to 1125 °C in the air for 3 h to find their highest density. The ceramics sintered at 1075 °C with the highest theoretical density (98.6%) were analyzed in further investigations. CaCu_3_Ti_4_O_12_ ceramics for comparison were prepared by the same experimental process.

The crystal structure of La_2/3_Cu_3_Ti_4_O_12_ ceramics was confirmed by X-ray diffraction (XRD, Bruker D8Advance, Billerica, MA, USA) with Rietveld analysis. The microstructure of samples (cleaned and broken in half) was observed by scanning electron microscopy (SEM, JSM-5610LV, JEOL, Tokyo, Japan) on their fractured surface. The valent state of different ions was analyzed by X-ray photoelectron spectroscopy (XPS, PHI-5000C ESCA, Perkin Elmer, Waltham, MA, USA) combining the XPSpeak4.1 fitting tool. Both sides of the samples were coated with silver paste. Turnkey Concept 50, the broadband dielectric spectrometer (Turnkey Concept 50, Novocontrol Technologies, Montabaur, Germany) was applied to measure the dielectric properties of samples in the frequency range of 1 Hz to 2 MHz from 123 to 600 K. The modified Debye equation is applied to analyze the frequency dependence of the dielectric constant. Arrhenius equations are used to analyze the temperature dependence of relaxation time and dc conductivity, respectively.

## 3. Results

The XRD data with the Rietveld refinement of La_2/3_Cu_3_Ti_4_O_12_ ceramics is shown in Figure 2. Almost all peaks are indexed, but one weak peak representing the secondary phase of TiO_2_ (1.39 wt %) was found. As shown in Table 1, La^3+^ ions are partially located in the *A* site for charge balance, and the lattice parameter of La_2/3_Cu_3_Ti_4_O_12_ ceramics with cubic structure is 7.418 Å. The increased lattice constant compared with CaCu_3_Ti_4_O_12_ ceramics (7.392 Å) may be due to the much larger La ions (*r*(La^3+^) = 1.061 Å > *r*(Ca^2+^) = 0.99 Å).

Figure 3 represents the SEM pictures of the cross-section for La_2/3_Cu_3_Ti_4_O_12_ and CaCu_3_Ti_4_O_12_ ceramics. Both the samples prepared by the same process were chosen with their highest densification. The average grain size of La_2/3_Cu_3_Ti_4_O_12_ ceramics is about 2 μm, and the complete substitution of rare earth elements La on Ca site in CaCu_3_Ti_4_O_12_ ceramics seems to improve the density of ceramics and refine the grain size to a certain extent.

The temperature dependence of dielectric properties for La_2/3_Cu_3_Ti_4_O_12_ ceramics is represented in Figure 4. Dielectric properties of La_2/3_Cu_3_Ti_4_O_12_ and CaCu_3_Ti_4_O_12_ ceramics at room temperature and different frequencies are listed in Table 2. The total substitution of La^3+^ for Ca^2+^ ions in CaCu_3_Ti_4_O_12_ ceramics has greatly increased the room temperature dielectric constant. Two dielectric relaxations are observed at two different temperature ranges (low-*T*: 123–200 K; high-*T* range 300–550 K), which is similar to those observed in CaCu_3_Ti_4_O_12_ ceramics. The dielectric constant ε′ decreases suddenly when the samples are cooled to a critically low temperature and the dielectric constant plateaus reduce slightly with increasing the frequency, while the dielectric peaks (about 6 × 10^4^) at high temperatures significantly decrease with increasing frequency. The low-*T* and high-*T* dielectric relaxations both show obvious frequency dispersion. Two parts of dielectric loss peaks are observed at the corresponding critical temperatures. However, two dielectric relaxations in La_2/3_Cu_3_Ti_4_O_12_ ceramics both move to a lower temperature (as shown in Figure 5).

Figure 6 represents the low-*T* dielectric properties of La_2/3_Cu_3_Ti_4_O_12_ ceramics as a function of frequency compared with CaCu_3_Ti_4_O_12_. The dielectric constant in CaCu_3_Ti_4_O_12_ ceramics exhibits better frequency stability than that of La_2/3_Cu_3_Ti_4_O_12_ ceramics as frequency decreases below about 1 kHz. However, the dielectric constant of both samples decreases abruptly as increasing the frequency above 1 kHz. Meanwhile, dielectric loss peaks associated with the abrupt change of dielectric constant show frequency dispersion characteristics, which can be well expressed with the modified Debye Equation.
(1)$$\varepsilon =\varepsilon^{\prime }-i\varepsilon^{\prime \prime }=\varepsilon_{\infty }+\left(\varepsilon_{s}-\varepsilon_{\infty }\right)/[1+\left(i\omega \tau {)}^{1-\alpha }\right]$$
where *ε_s_* and *ε_∞_* are static and the infinite frequency dielectric constant, *ω* represents the angular frequency, *τ* is the mean relaxation time, and α is the distribution of relaxation time. The temperature dependence of *τ* can be fitted with the Arrhenius equation
(2)$$\tau =\tau_{0}exp(E_{\mathrm{relax}}/k_{B}T)$$

*τ*_0_, *E*_relax_ and *k*_B_ represented the prefactor, activation energy, and the Boltzmann constant, respectively. The activation energy for low-*T* dielectric relaxation of La_2/3_Cu_3_Ti_4_O_12_ ceramics (*E*_a_ = 0.087 eV) is smaller than that of CaCu_3_Ti_4_O_12_ ceramics (*E*_a_ = 0.125 eV), which result in a lower critical temperature and is easier for inducing dielectric response. Moreover, the decrease in activation energy for low-*T* dielectric relaxation may indicate the stronger grain effect related to increased dipoles from polyvalent Ti and Cu ions.

XPS was applied to further investigate the electronic configuration of La_2/3_Cu_3_Ti_4_O_12_. Figure 7 shows the XPS spectra of Ti and Cu ions in La_2/3_Cu_3_Ti_4_O_12_ ceramics compared with CaCu_3_Ti_4_O_12_ ceramics. According to the NIST XPS database and Gaussian-Lorentzian fitting method, the 2p_3/2_ peaks for Ti and Cu ions are split into two peaks, indicating the coexistence of variable ions with low states such as Ti^3+^ and Cu^+^. Both the area ratios related to the atomic ratio (Ti^3+^/Ti^4+^, Cu^+^/Cu^2+^) for La_2/3_Cu_3_Ti_4_O_12_ ceramics increase compared with that for CaCu_3_Ti_4_O_12_ ceramics as shown in Table 3. The increased concentration of Cu^+^ and Ti^3+^ ions in La_2/3_Cu_3_Ti_4_O_12_ ceramics may produce more dipoles, thus reducing the activation energy of low-temperature relaxation.

The frequency dependence of the high-*T* dielectric constant in La_2/3_Cu_3_Ti_4_O_12_ ceramics from 357 K to 505 K is shown in Figure 8, and the fitting result with Equations (1) and (2). The activation energy *E*_relax_ for high-*T* dielectric relaxation is 0.596 eV close to the those reported values in other *A*Cu_3_Ti_4_O_12_ (*A* = Ca, Dy_2/3_ et al.) compounds, which can be related to the grain boundary barrier effect.

Figure 9 represents the frequency dependence of ac conductivity of La_2/3_Cu_3_Ti_4_O_12_ ceramics in a high-temperature range from 357 to 505 K. The extrapolated dc conductivity at a low frequency well obeys the Arrhenius relation [18] as a function of temperature (as shown in the inset of Figure 9). The calculated activation energy of conductivity (*E*_dc_ = 0.571 eV) is similar to the previously reported value for La_2/3_Cu_3_Ti_4_O_12_ ceramics [26,27,28], and is close to that of high-*T* dielectric relaxation (*E*_relax_ = 0.596 eV) and also implies the correlation between dielectric relaxation and conduction at high temperature.

According to IBLC electrical model, the imaginary part of impedance Z″ can be expressed as a function of frequency.
(3)$$Z_{\prime \prime }(\omega )=R_{g}\left[\frac{\omega R_{g}C_{g}}{1+{\left(\omega R_{g}C_{g}\right)}^{2}}\right]+R_{gb}\left[\frac{\omega R_{gb}C_{gb}}{1+{\left(\omega R_{gb}C_{gb}\right)}^{2}}\right]$$
where *R_g_* and *R_gb_* are the resistance of grains and grain boundaries, *C_g_* and *C_gb_* are the capacitance of grains and grain boundaries, respectively. *Z*″ peaks in Figure 10 indicate a thermally activated electrical property of grain boundaries according to the equation *R_gb_* ≈ 2Z″ max [14]. The shift of the *Z*″ peak towards higher frequencies with the increasing temperature well follows the Arrhenius law [18] with the conductive activation energy as 0.598 eV in La_2/3_Cu_3_Ti_4_O_12_ ceramics. The activation energy for conduction is close to that of the high-*T* dielectric relaxation, which may imply that conductivity and dielectric relaxation in La_2/3_Cu_3_Ti_4_O_12_ ceramics at high temperature have similar mechanisms due to the grain boundaries effects. Compared with CaCu_3_Ti_4_O_12_ ceramics (*E*_gb_ = 0.639 eV), the reduction of high-*T* activation energies for both conductivity and dielectric relaxation in La_2/3_Cu_3_Ti_4_O_12_ ceramics can be associated with increased defects (such as oxygen vacancy, polyvalent ions) mainly due to the charge compensation effect from the complete substitution of La^3+^ ions on Ca site in CaCu_3_Ti_4_O_12_ ceramics. The detailed mechanism still needs further investigation.

## 4. Conclusions

La_2/3_Cu_3_Ti_4_O_12_ and CaCu_3_Ti_4_O_12_ ceramics were prepared by the same process of solid-state reaction. The similarity and difference in structure, dielectric properties, and relative mechanism in these ceramics were comparatively investigated. The similarities are that La_2/3_Cu_3_Ti_4_O_12_ ceramics exhibit a similar high dielectric constant (*ε*′~10^4^) and two distinct dielectric relaxations to CaCu_3_Ti_4_O_12_ ceramics. The dielectric relaxation below 200 K with an activation energy of 0.087 eV in La_2/3_Cu_3_Ti_4_O_12_ ceramics is due to the polyvalent state of Ti^3+^/Ti^4+^ and Cu^+^/Cu^2+^, while the dielectric relaxation above 450 K with higher activation energy (0.596 eV) is due to grain boundary effects. The origin of the giant dielectric constant in La_2/3_Cu_3_Ti_4_O_12_ ceramics can be also explained by the IBLC mechanism. Meanwhile, the differences in structure are that the density has been increased and the grain size has been refined in La_2/3_Cu_3_Ti_4_O_12_ ceramics, which can be due to the introduction of La rare earth elements (proved by other groups [24]). Moreover, the two thermal activated dielectric relaxations with increased dielectric constant and decreased activation energy in La_2/3_Cu_3_Ti_4_O_12_ ceramics both move to lower temperatures, which can be related to the enhanced defect structure. The heterovalent ion substitution of La^3+^ on Ca^2+^ ions can induce more ions to get electrons for share conservation, which will increase the lower valence state of Ti^3+^ and Cu^+^ ions that could produce more dipoles and defects, and as the result decrease the activation energy of dielectric relaxation.

## Figures and Tables

**Figure 1 materials-15-04526-f001:**
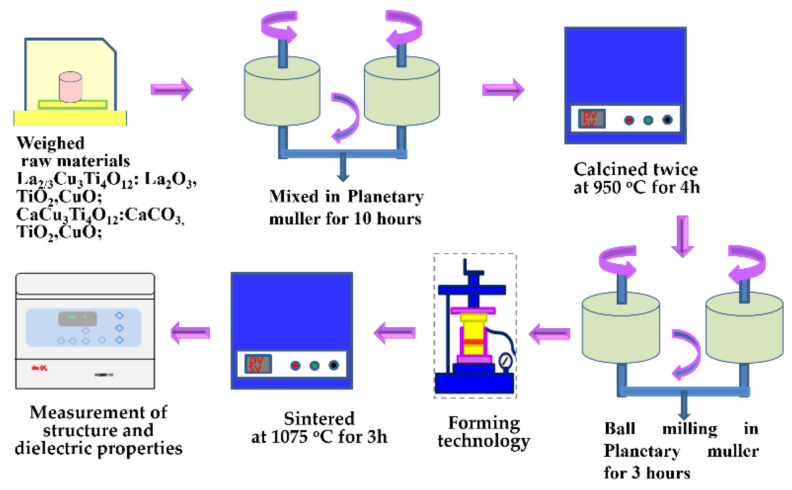
Flow chart of sample preparation.

**Figure 2 materials-15-04526-f002:**
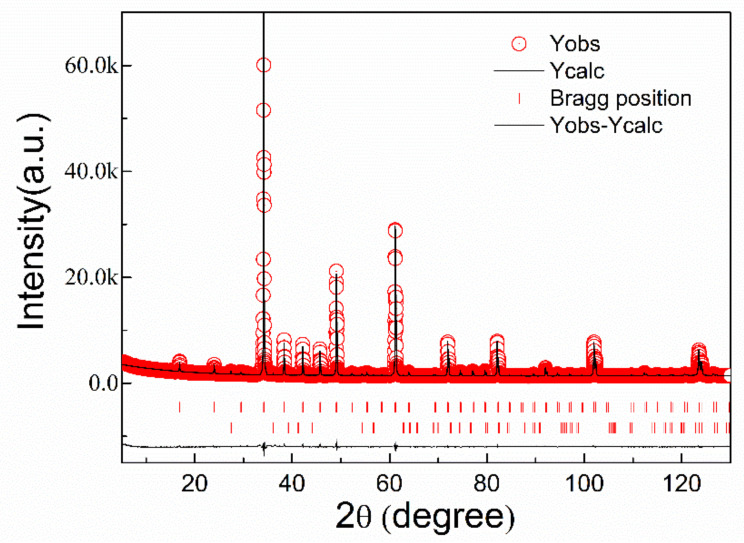
XRD pattern with Rietveld analysis of La_2/3_Cu_3_Ti_4_O_12_ ceramics: *Y*_obs_ is experiment data. *Y*_cal_ is the calculation results. Two short vertical lines below represent La_2/3_Cu_3_Ti_4_O_12_ ceramics and a minor amount of second phase TiO_2_ (1.39 wt %), respectively.

**Figure 3 materials-15-04526-f003:**
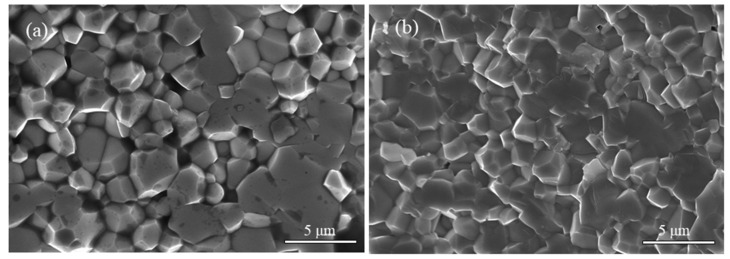
SEM pictures of the cross-section for CaCu_3_Ti_4_O_12_ (**a**) and La_2/3_Cu_3_Ti_4_O_12_ (**b**) ceramics.

**Figure 4 materials-15-04526-f004:**
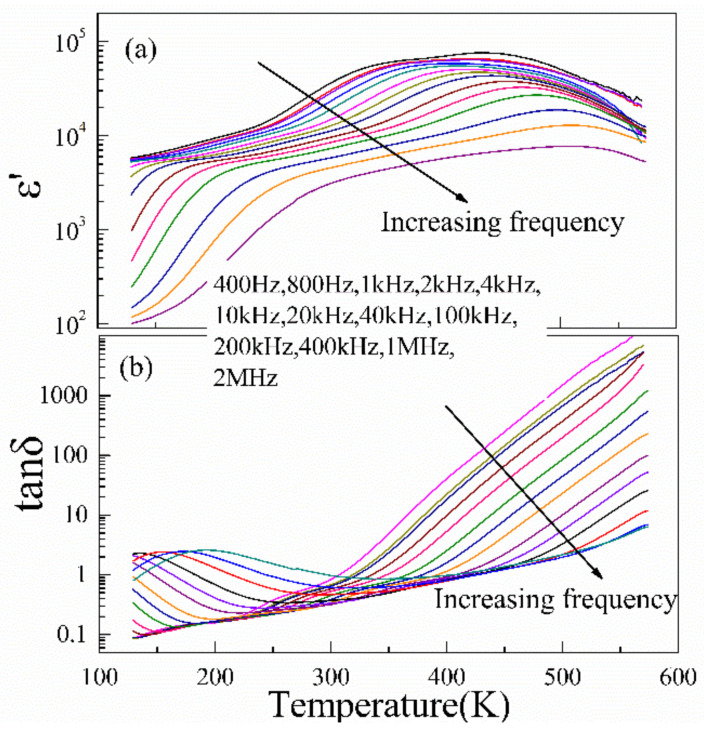
Temperature dependence of dielectric constant (**a**) and dielectric loss (**b**) for La_2/3_Cu_3_Ti_4_O_12_ ceramics at various frequencies (400 Hz–2 MHz).

**Figure 5 materials-15-04526-f005:**
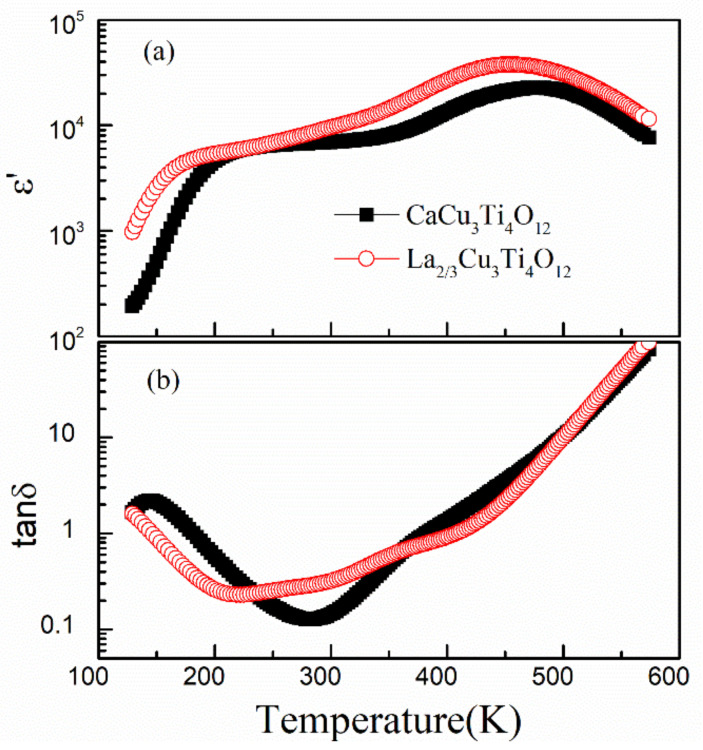
Temperature dependence of the dielectric properties of La_2/3_Cu_3_Ti_4_O_12_ and CaCu_3_Ti_4_O_12_ ceramics at 100 kHz. (**a**) Dielectric constant and (**b**) dielectric loss.

**Figure 6 materials-15-04526-f006:**
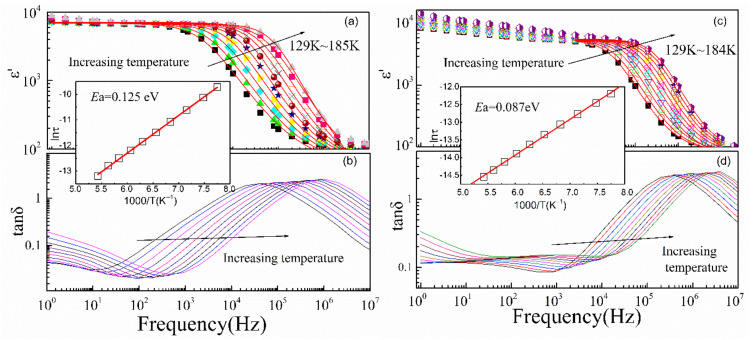
Frequency dependence of dielectric properties of CaCu_3_Ti_4_O_12_ (**a**,**b**) and La_2/3_Cu_3_Ti_4_O_12_ (**c**,**d**) ceramics, respectively, in a low-temperature range (129–184 K). In figures (**a**,**c**), the experimental data are represented by the solid symbols, and the fitting results by the modified Debye’s model are the lines. Insets show the temperature dependence of the relaxation time *τ*.

**Figure 7 materials-15-04526-f007:**
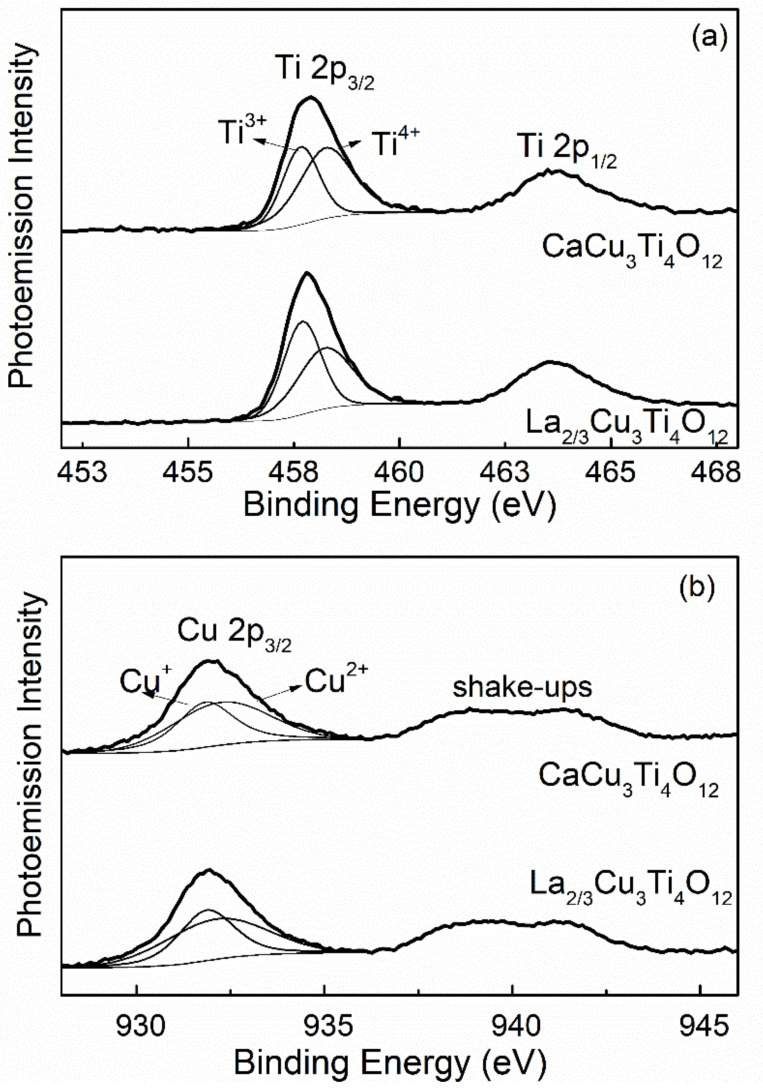
XPS of the Ti (**a**) and Cu (**b**) ions of La_2/3_Cu_3_Ti_4_O_12_ and CaCu_3_Ti_4_O_12_ ceramics, indicating the polyvalence Ti^3+^/Ti^4+^ and Cu^+^/Cu^2+^.

**Figure 8 materials-15-04526-f008:**
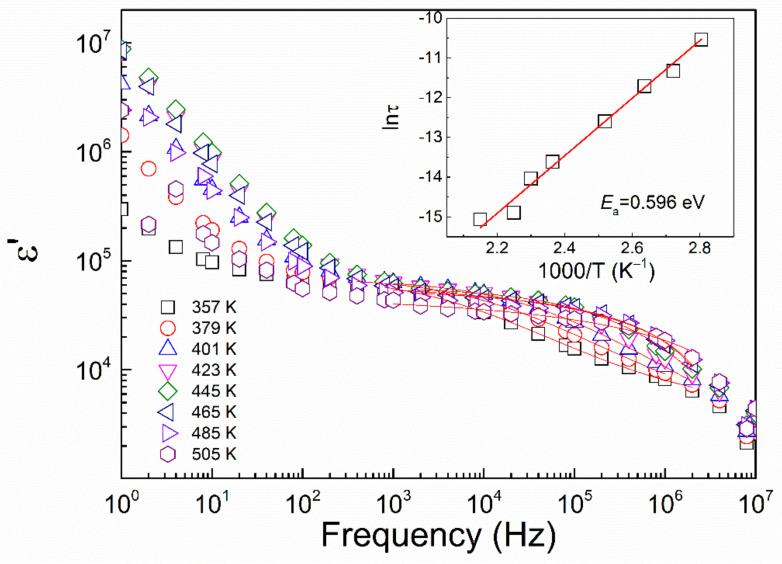
Frequency dependence of high-*T* dielectric constant in La_2/3_Cu_3_Ti_4_O_12_ ceramics from 357 K to 505 K. Experimental data are represented by solid symbols which are linearly fitted by Equation (1). Inset represents the linear dependence of the relaxation time *τ* vs. temperature.

**Figure 9 materials-15-04526-f009:**
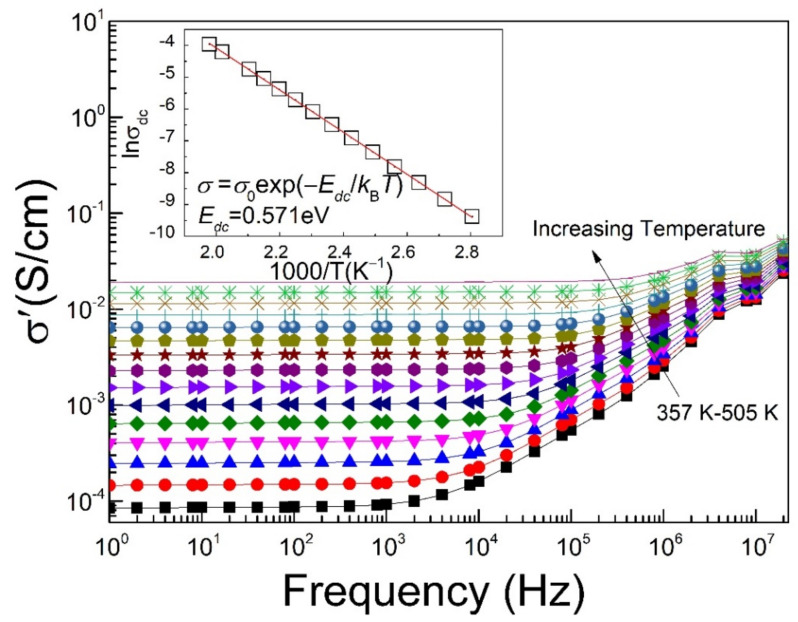
Frequency dependence of ac conductivity in La_2/3_Cu_3_Ti_4_O_12_ ceramics from 357 K to 505 K. The inset shows the linear dependence of extrapolated dc conductivity with temperature.

**Figure 10 materials-15-04526-f010:**
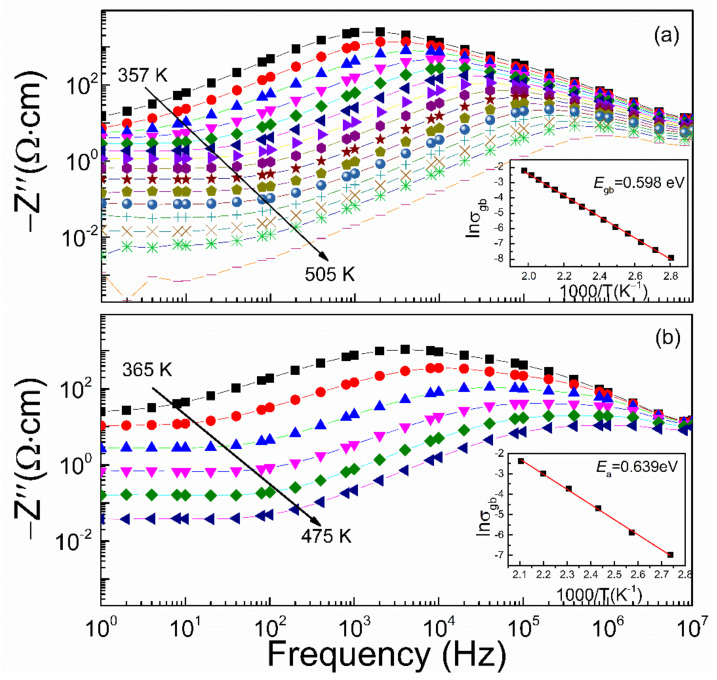
Frequency dependence of *Z*″ for La_2/3_Cu_3_Ti_4_O_12_ (**a**) and CaCu_3_Ti_4_O_12_ (**b**) in the high-*T* range. Solid symbols in insets are experimental data representing the temperature dependence of grain boundary conductivity σ_gb_, and the fitting results are shown as red lines.

**Table 1 materials-15-04526-t001:** Structure Parameters for La_2/3_Cu_3_Ti_4_O_12_ ceramics.

Atom	Wyckoff Prosition	x	y	z	Biso(Å^2^)	Occupies
La	2a	0.000	0.000	0.000	0.021(39)	0.028(0)
Cu	6b	0.000	0.500	0.500	0.373(29)	0.125(0)
Ti	8c	0.250	0.250	0.250	0.499(26)	0.167(0)
O	24g	0.30336	0.18024	0.000	0.030(0)	0.500(0)

Lattice parameters: *a* = 7.41773 Å, *V* = 408.144 Å^3^, space group Im3¯ (204). Agreement indices: *R*_p_ = 2.68, *R*_wp_ = 3.48, *χ*^2^ = 2.2.

**Table 2 materials-15-04526-t002:** Dielectric properties of La_2/3_Cu_3_Ti_4_O_12_ and CaCu_3_Ti_4_O_12_ ceramics at 298 K and different frequencies.

Compound	1 kHz		100 kHz	
	*ε*′	tan*δ*	*ε*′	tan*δ*
La_2/3_Cu_3_Ti_4_O_12_	27,753	0.63	9,396.5	0.31
CaCu_3_Ti_4_O_12_	13,761	0.90	6,904.2	0.14

**Table 3 materials-15-04526-t003:** XPS parameters of CaCu_3_Ti_4_O_12_ and La_2/3_Cu_3_Ti_4_O_12_ ceramics.

Compound	Bingding Energy (eV)				Area Ratio	
	Cu^+^	Cu^2+^	Ti^3+^	Ti^4+^	Cu^+^/Cu^2+^	Ti^3+^/Ti^4+^
CaCu_3_Ti_4_O_12_	931.826	932.232	457.657	458.261	0.799	0.686
La_2/3_Cu_3_Ti_4_O_12_	931.864	932.108	457.691	458.233	0.822	1.007

## Data Availability

The article includes all data.

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
