# Peer review of "High Dielectric Constant and Dielectric Relaxations in La2/3Cu3Ti4O12 Ceramics"

_materials, 2022, doi:10.3390/ma15134526_

Round 1

Reviewer 1 Report

This is a nice paper presenting the structure and dielectric responses for La2/3Cu3Ti4O12 and CaCu3Ti4O12 ceramics. Overall, the manuscript is written in clear English. The paper may be published in the Materials journal, however, after some important corrections, as justified in the following points:

1) The electrical contact is very important for evaluation of dielectric properties of the prepared materials. How the authors prepared the electrodes on both sides of the samples?

2) The conductivity is an important aspect that should be taken into consideration for practical use. Therefore, I recommend authors to include the frequency-dependent conductivity spectra. Alternatively, the impedance plots -Z″ vs. Z′ for La2/3Cu3Ti4O12 ceramics may be presented. An example of the evaluation of dc-conductivity may be found in ACS Applied Polymer Materials 3, 4869−4878 (2021). Hence, a complete examination based on dielectric spectroscopy measurements will be achieved.

3) The paper is clear, reads well and brings a lot of interesting experimental data. However, for a close view of the results, the numerical values of the dielectric parameters may be included into a Table.

Author Response

Dear Professor:

Thank you very much for your kind reviewing and valuable comments on my manuscript entitled “High dielectric constant and dielectric relaxations in La2/3Cu3Ti4O12 ceramics” (No. Materials-1755069) by L. Ni, C.Y. Zhang and L. Fang. I have substantially modified the manuscript according to your comments.

Response to the reviewer 1:

Point 1: The electrical contact is very important for evaluation of dielectric properties of the prepared materials. How the authors prepared the electrodes on both sides of the samples?

Response 1: Both sides of the samples were coated with silver paste. The corresponding description has been modified in the revised manuscript.

Point 2: The conductivity is an important aspect that should be taken into consideration for practical use. Therefore, I recommend authors to include the frequency-dependent conductivity spectra. Alternatively, the impedance plots -Z″ vs. Z′ for La2/3Cu3Ti4O12 ceramics may be presented. An example of the evaluation of dc-conductivity may be found in ACS Applied Polymer Materials 3, 4869−4878 (2021). Hence, a complete examination based on dielectric spectroscopy measurements will be achieved.

Response 2: The frequency dependence of ac conductivity of La2/3Cu3Ti4O12 ceramics at high temperature has been added in the revised manuscript, and the extrapolated dc conductivity at low frequency well obeys the Arrhenius relation as function of temperature. The calculated activation energy of conductivity is quite close to that of high-T dielectric relaxation, which implies the correlation between dielectric relaxation and conduction at high temperature. The corresponding description has been also modified.

Point 3: The paper is clear, reads well and brings a lot of interesting experimental data. However, for a close view of the results, the numerical values of the dielectric parameters may be included into a Table.

Response 3: Table 2 represented the dielectric properties of La2/3Cu3Ti4O12 and CaCu3Ti4O12ceramics has been added in the revised manuscript. The corresponding description has been also modified. 

I hope the above modifications can satisfy your standard, and I would much appreciate your further reviewing and comments.

Reviewer 2 Report

1. Please highlight the list of analyses used in this work in the abstract section.

2. Please state the potential application in the last line of the abstract section.

3. In the introduction section, the author needs to explain how to fill the knowledge gap and lay out the research objectives and methodology.

4. Please state the problem statement and objective of the work.

5. In the methodology section, the author needs to explain comprehensively

I. Describe the materials and equipment

ii. Explain how samples were gathered

iii. Explain how measurements were made

iv. Describe the calculations that were performed on the data

v. Describe the statistical techniques used

6. Simple flow chart of sample preparation was compulsory to be added in the text to describe the whole research work process.

7. Several related previous works need to be added to the discussion part.

i. Ahmad, M. M., Kotb, H. M., Joseph, C., Kumar, S., & Alshoaibi, A. (2021). Transport and Dielectric Properties of Mechanosynthesized La2/3Cu3Ti4O12 Ceramics. Crystals, 11(3), 313.

ii. Fu, Z., Nie, H., Wei, Y., Zhang, B., & Chang, A. (2020). Effect of Mn-doping on microstructure and electrical properties of La2/3Cu3Ti4O12 ceramics. Journal of Alloys and Compounds, 847, 156525.

iii. Ismail, I., Ibrahim, I.R., Matori, K.A., Awang, Z., Zulkimi, M.M.M., Idris, F.M., Nazlan, R., Zaid, M.H.M., Rusly, S.N.A. and Ertugrul, M., 2020. Comparative study of single-and double-layer BaFe12O19-Graphite nanocomposites for electromagnetic wave absorber applications. Materials Research Bulletin, 126, p.110843.

8. Please improve the quality of the information in the graph, figures, and tables.

9. Please position your findings in the context of previous research; do they contradict or add something new. Some references maybe can be added to the text to support the statement.

10. In the conclusion part, please provide a short synopsis of the results and discussion, summing up the paper.

11. Some recent related publications (2020, 2021, and 2022) must be added as references.

Author Response

Thank you very much for your kind reviewing and valuable comments on my manuscript entitled “High dielectric constant and dielectric relaxations in La2/3Cu3Ti4O12 ceramics” (No. Materials-1755069) by L. Ni, C.Y. Zhang and L. Fang. I have substantially modified the manuscript according to your comments.

 Response to the reviewer 2:

Point 1: Please highlight the list of analyses used in this work in the abstract section.

Response 1: The list of analyses used in this work has been highlighted in the abstract section. “The structure and dielectric responses for La2/3Cu3Ti4O12 and CaCu3Ti4O12 ceramics were systematically investigated by x-ray diffraction, scanning electron microscopy, x-ray photoelectron spectroscopy and impedance analyzer et.al.”

Point 2: Please state the potential application in the last line of the abstract section.

Response 2: The potential application has been added in the last line of the abstract section. “Such high dielectric constant ceramics are also expected to be applied in capacitors and memory devises et. al.

Point 3: In the introduction section, the author needs to explain how to fill the knowledge gap and lay out the research objectives and methodology.

Response 3: The introduction has been revised. The explanation about the knowledge gap, the research objectives and methodology has been added in the revised manuscript.

Until now, there has been a certain amount of work on La2/3Cu3Ti4O12 ceramics, which mainly focused on the effects of fabrication conditions and doping on the structure and properties of La2/3Cu3Ti4O12 ceramics. However, there are few comparative studies on the dielectric relaxation and relative mechanism of La2/3Cu3Ti4O12 and CaCu3Ti4O12 ceramics with similar sintering parameters. Moreover, there are still some questions unsolved, such as, what are the changes in structure and dielectric relaxation caused by the complete substitution of hetero-valent ions on A site in ACu3Ti4O12 ceramics. In order to comparatively investigate the similarity and difference of dielectric relaxations in La2/3Cu3Ti4O12 and CaCu3Ti4O12 ceramics, the same process was used to prepare La2/3Cu3Ti4O12 and CaCu3Ti4O12 ceramics, and the structure and dielectric properties of La2/3Cu3Ti4O12 and CaCu3Ti4O12 ceramics were systematically investigated by using x-ray diffraction, scanning electron microscopy, x-ray photoelectron spectroscopy and impedance analyzer et.al.

Point 4: Please state the problem statement and objective of the work.

Response 4: As mentioned above in response 3, Although there has been a certain amount of work focused on the effects of fabrication conditions and doping on the structure and properties in La2/3Cu3Ti4O12 ceramics, there are few comparative studies on the dielectric relaxation and relative mechanism of La2/3Cu3Ti4O12 and CaCu3Ti4O12 ceramics with similar sintering parameters. Moreover, there are still some questions unsolved, such as, what are the changes in dielectric relaxation and giant dielectric plateaus caused by the complete substitution of hetero-valent ions on A site in ACu3Ti4O12 ceramics.

    The objective of the work is to comparatively investigate the similarity and difference of dielectric relaxations in La2/3Cu3Ti4O12 and CaCu3Ti4O12 ceramics.

Point 5: In the methodology section, the author needs to explain comprehensively

  1. Describe the materials and equipment
  2. Explain how samples were gathered

iii. Explain how measurements were made

  1. Describe the calculations that were performed on the data
  2. Describe the statistical techniques used

Response 5: Affected by the article’s repetition rate, the methodology section had been simply described in this manuscript. The five suggestions above have been adopted. Some have been modified in the revised manuscript, and the rest of them are explained as follows:

  1. Describe the materials and equipment: modified in the revised manuscript.
  2. Explain how samples were gathered:

Raw materials weighed and mixed in planetary muller for 10 hours, were then heated twice at 950 oC for 4h. The powders mixed with 8% PVA were pressed into disks (thickness: 2mm; diameter:12 mm) under a pressure of 98MPa. The disks (about 5 disks per sintering temperature) were sintered from 1050 oC to 1125 oC in air for 3 h to find its highest density (>98% of the theoretical density).

iii. Explain how measurements were made:

The detail measurements have been partially added in the revised manuscript. The following description is a more detailed supplementary illustration.

The crystal phase of the sintered ceramics (grounded to a fine powder in an agate mortar) was confirmed by powder X-ray diffraction using Cu Ka radiation. The XRD data for Rietveld analysis were collected over the range of 2q=15-130o with a step size of 0.02o and a count time of 2s. The Rietveld refinement was performed using the FULLPROF program, and a pseudo-Voigt profile function with preferred orientation was used.

The microstructure of the samples (cleaned and broke in half) was evaluated by scanning electron microscopy on their fractured surface.

The samples were grinded for x-ray photoelectron spectroscopy analysis with Mg Ka radiation (hn=1253.6 eV). The experimental curve was fitted with a program (XPSPeak4.1) using a combination of Gaussian-Lorentzian lines.

The surface of the samples was polished on metallographic sandpaper. The silver electrodes were obtained by coating silver paste and cofiring at 823 K for 30 minutes in air. The dielectric characteristics of the samples were evaluated with a broadband dielectric spectrometer. The final values of the dielectric properties were derived from the average of three samples produced in the same batch.

  1. Describe the calculations that were performed on the data: modified in the revised manuscript.

“The modified Debye equation is applied to analyze the frequency dependence of dielectric constant. Arrhenius equations are used to analyze the temperature dependence of relaxation time and dc conductivity respectively.”

  1. Describe the statistical techniques used

The XRD data for Rietveld analysis were collected over the range of 2q=15-130o with a step size of 0.02o and a count time of 2s. The Rietveld refinement was performed using the FULLPROF program, and a pseudo-Voigt profile function with preferred orientation was used. The experimental curve of XPS analysis was fitted with a program (XPSPeak4.1) using a combination of Gaussian-Lorentzian lines. The final values of the dielectric properties were derived from the average of three samples produced in the same batch. The frequency dependence of dielectric constant was fitted by the modified Debye equation. Temperature dependence of relaxation time and dc conductivity was fitted by Arrhenius equation.

Point 6: Simple flow chart of sample preparation was compulsory to be added in the text to describe the whole research work process.

Response 6: The flow chart of sample preparation as Figure 1 has been added in the revised manuscript.

Point 7: Several related previous works need to be added to the discussion part.

Response 7: The closely related works have been added to the discussion part. “The calculated activation energy of conductivity (Edc = 0.571eV) is similar to the previous reported value for La2/3Cu3Ti4O12 ceramics [26-28]”

Point 8: Please improve the quality of the information in the graph, figures, and tables.

Response 8: The figures with obvious quality problems have been modified and exported from OriginPro 9.0 with the 600 resolutions, such as Figure 4, 6,7,8, and 10.

Point 9: Please position your findings in the context of previous research; do they contradict or add something new. Some references maybe can be added to the text to support the statement.

Response 9: The findings in our work (especially the conductive mechanism and dielectric relaxation in the high temperature) do not contradict with the previous research, and some relative references have been added to the text for support. The innovation of our work is the changes in dielectric relaxations and the giant dielectric plateaus of La2/3Cu3Ti4O12 ceramics compared with CaCu3Ti4O12 ceramics under the condition of the similar preparation process.

Point 10: In the conclusion part, please provide a short synopsis of the results and discussion, summing up the paper.

Response 10: The conclusion part has been modified in the revised manuscript.

La2/3Cu3Ti4O12 and CaCu3Ti4O12 ceramics were prepared by the same process of solid-state reaction. The similarity and difference of structure, dielectric properties and relative mechanism in these ceramics were comparatively investigated. The similarities are that La2/3Cu3Ti4O12 ceramics exhibit the similar high dielectric constant (e’ ~ 104) and two distinct dielectric relaxations to CaCu3Ti4O12 ceramics. The dielectric relaxation below 200 K with activation energy 0.087 eV in La2/3Cu3Ti4O12 ceramics is due to the polyvalent state of Ti3+/Ti4+ and Cu+/Cu2+, while the dielectric relaxation above 450K with higher activation energy (0.596 eV) is due to grain boundary effects. The origin of giant dielectric constant in La2/3Cu3Ti4O12 ceramics can be also explained by the IBLC mechanism. Meanwhile, the differences in structure are that the density has been increased and the grain size has been refined in La2/3Cu3Ti4O12 ceramics, which can be due to introduction of La rare earth elements (proved by other groups). Moreover, the two thermal activated dielectric relaxations with increased dielectric constant and decreased activation energy in La2/3Cu3Ti4O12 ceramics both move to the lower temperatures, which can be related to the enhanced defect structure. The hetero-valent ions substitution of La3+ on Ca2+ ions can induce more ions to get electron for chare conservation, which will increase the lower valence state Ti3+ and Cu+ ions that could produce more dipoles and defects, and as the result decrease the activation energy of dielectric relaxation.

Point 11: Some recent related publications (2020, 2021, and 2022) must be added as references.

Response 11: Some related works published recently (listed below) have been added as references in the revised manuscript.

13  Guo, Y.; Tan, J.L.; Zhao, J.C. Microstructure and electrical properties of nano-scale SnO2 hydrothermally coated CCTO-based composite ceramics. Ceram. Int. 2022, 48, 17795-17801.

15   Peng, Z,H,; Wang, J.T; Zhang, F.D.; Xu, S.D.; Lei, X.P.; Liang P.F.; Wei, Wu, D.; L.L.; Chao, X.L.; Yang, Z.P. High energy storage and colossal permittivity CdCu3Ti4O12 oxide ceramics. Ceram. Int. 2022, 48, 4255-4260.

17   Peng, Z,H,; Wang, X.; Xu, S.D.; Zhang, F.D.; Wang, J.T; Wang, J.J.; Wu, D.; Liang P.F.; Wei, L.L.; Chao, X.L.; Yang, Z.P. Improved grain boundary resistance inducing decreased dielectric loss and colossal permittivity in Y2/3Cu3Ti4O12 ceramics. Mater. Chem. Phys. 2022, 283, 125874.

25   Ahmad, M.M.; Kotb, H.M.; Joseph, C.; Kumar, S.; Alshoaibi, A. Transport and dielectric properties of mechanosynthesized La2/3Cu3Ti4O12 ceramics. Crystals 2021, 11 ,313.

26   Fu, Z.L.; Nie, H.C.; Wei, Y.X.; Zhang, B.; Chang, A.M. Effect of Mn-doping on microstructure and electrical properties of La2/3Cu3Ti4O12 ceramics. J. Alloy. Compd. 2020, 847, 156525.

27   Liu, Z.Q. Dielectric properties and impedance versus dc bias and I-V characteristics of La2/3Cu3Ti4O12 ceramics. Ferroelectrics 2020 555, 199-210.

I hope the above modifications can satisfy your standard, and I would much appreciate your further reviewing and comments.

Round 2

Reviewer 1 Report

I agree with the revision version of the manuscript.

Reviewer 2 Report

All the correction has been done by the author. The revised version manuscript can be accepted.